# Therapeutic Associations Comprising Anti-PD-1/PD-L1 in Breast Cancer: Clinical Challenges and Perspectives

**DOI:** 10.3390/cancers13235999

**Published:** 2021-11-29

**Authors:** Fanny Ledys, Laura Kalfeist, Loick Galland, Emeric Limagne, Sylvain Ladoire

**Affiliations:** 1Platform of Transfer in Cancer Biology, Georges-François Leclerc Center, 21000 Dijon, France; fledys@cgfl.fr (F.L.); lkalfeist@cgfl.fr (L.K.); lgalland@cgfl.fr (L.G.); elimagne@cgfl.fr (E.L.); 2School of Medicine and Pharmacy, University of Burgundy Franche-Comté, 21000 Dijon, France; 3UMR INSERM 1231, Lipides Nutrition Cancer, 21000 Dijon, France; 4Department of Medical Oncology, Georges-François Leclerc Center, 21000 Dijon, France

**Keywords:** breast cancer, PD-1/PD-L1 blockade, immune response, chemotherapies, kinase inhibitors, targeted therapies

## Abstract

**Simple Summary:**

Breast cancer remains the leading cause of death in women. Despite improved treatment in recent years, new therapeutic options are still needed for some types of breast cancer. In view of the poor response of breast cancer to immunotherapy, it is important to develop therapeutic combinations in order to sensitize breast tumors to anti-PD-(L)1 immunotherapy. This review presents the different combinations developed in pre-clinical and clinical studies according to the immune characterization of breast cancers.

**Abstract:**

Despite a few cases of long-responder patients, immunotherapy with anti-PD-(L)1 has so far proved rather disappointing in monotherapy in metastatic breast cancer, prompting the use of synergistic therapeutic combinations incorporating immunotherapy by immune-checkpoint inhibitors. In addition, a better understanding of both the mechanisms of sensitivity and resistance to immunotherapy, as well as the immunological effects of the usual treatments for breast cancer, make it possible to rationally consider this type of therapeutic combination. For several years, certain treatments, commonly used to treat patients with breast cancer, have shown that in addition to their direct cytotoxic effects, they may have an impact on the tumor immune microenvironment, by increasing the antigenicity and/or immunogenicity of a “cold” tumor, targeting the immunosuppressive microenvironment or counteracting the immune-exclusion profile. This review focuses on preclinical immunologic synergic mechanisms of various standard therapeutic approaches with anti-PD-(L)1, and discusses the potential clinical use of anti-PD-1/L1 combinations in metastatic or early breast cancer.

## 1. Background and Current Development of Immunotherapy with Anti-PD-1/PD-L1 in Breast Cancer

According to the latest global cancer statistics (GLOBOCAN 2020), breast cancer (BC) ranks among the most frequently encountered cancers, with a rate of 11.7%. BC accounted for 6.9% of cancer deaths in 2020, making it the leading cause of cancer death in women worldwide [1]. BC has traditionally been classified into three subtypes associated with different prognosis and substantial microenvironment heterogeneity. Among these subsets, luminal cancers are characterized by estrogen (ER) +/− progesterone receptor (PR) expression. Conventional systemic treatment (both in the adjuvant and metastatic setting) is based on hormone therapy, and prognosis is relatively good. The second type, called HER2-positive BC, is characterized by high expression of this receptor and does not express hormone receptors, and is treated with HER2-targeted therapies, most often in combination with chemotherapy. Finally, the third type is triple-negative BC (TNBC), which is characterized by the absence of expression of any of the three predictive markers (i.e., ER, PR or HER2). Therefore, there is currently no specific targeted therapy for this subtype, and chemotherapy remains the standard of systemic treatment, always in the adjuvant or metastatic setting. The latter two types of cancer have a poor prognosis [2]. 

Tumor-mediated immune escape is one of the hallmarks of cancer that blunts cancer cell detection and elimination by cytotoxic lymphocytes. This phenomenon implicates a large number of biological processes, and one of them is immune checkpoint induction, such as cytotoxic T-lymphocyte-associated protein 4 (CTLA-4) or PD-1, which have suppressive functions against activation of anti-tumor lymphocytes. Blockade of these immune checkpoints by monoclonal antibodies has positively impacted the therapeutic management of a broad number of malignancies, by enabling profound and lasting tumor responses to be obtained, albeit in a small subset of patients [3].

Historically, BC was not considered to be immunologically active, especially compared to other tumors such as melanoma, renal or lung cancers [4,5], most often due to an “immunologically cold” contexture, associating low immune cell infiltration (in particular, cytotoxic T lymphocytes), and low tumor mutational burden (TMB). This is especially true for HER2 negative luminal tumors, which account for the overwhelming majority of breast cancers [6].

Nevertheless, there is mounting evidence that some BCs can present a more favorable immune biology, with a high level of tumor infiltrating lymphocytes (TILs). This infiltration is associated with better prognosis and also has a positive predictive impact on chemosensitivity, particularly in the context of neoadjuvant chemotherapy, where higher TIL level is associated with higher probability of pathological complete response (pCR) [7,8]. However, this prognostic and predictive impact of the efficacy of chemotherapy depends on the breast cancer subtype, and is particularly marked in the triple negative and HER2 positive subtypes, in which there is more often a higher TMB, as well as greater inflammatory signature expression. Moreover, in the HER2 positive subtype, the immune response (in particular TIL levels) also seems to influence the effectiveness of anti-HER2 treatments, in particular monoclonal antibodies (possibly by the phenomenon of Antibody-Dependent Cell Cytotoxicity (ADCC)) [7,9,10,11]. Among TILs, the presence of CD8^+^ T cells, as well as the ratio between effector CD8^+^ and regulatory FoxP3^+^ T cells, appears to be associated with both better prognosis and long-term survival [7,12,13]. T cells infiltration is variable between different breast cancer subtypes. Indeed, TNBC and HER2 positive tumors appear to be the most CD8^+^ T cells infiltrated with a median percentage of patients with high CD8^+^ T cells levels of 60% and 61% respectively compared to the RH+ subtype with 43%. Furthermore, 70% and 67% of TNBC and HER2 patients had high levels of FoxP3+ T cells compared to 38% of HR+ patients [14,15].

These observations suggest that therapeutic strategies based on cytotoxic immune cell stimulation and/or immunosuppression blockade can be effective for BC treatment. Among these strategies, the discovery of immune checkpoints, which regulate immune activation, and their blockade with monoclonal antibodies targeting PD-1/PD-L1, has been considered and tested.

The first studies were carried out with anti PD-(L)1 monotherapy in patients with advanced metastatic disease, and TNBC (supposed to be more immunogenic). Initial studies carried out with atezolizumab (anti PD-L1) or pembrolizumab (anti-PD-1) as monotherapy and in advanced stage, found objective response rates (ORR) around 20% (including in patients with PD-L1 negative tumors), with long-lasting tumor responses in some patients [16,17]. The mechanisms behind these very low response rates are still largely unknown. Like other tumor types, certain breast cancer tumor cells are, for example, capable of co-expressing PD-1 and PD-L1, which slows down tumor growth. Targeting PD-1 or PD-L1 is then paradoxically able to stimulate tumor growth. Whether this mechanism could explain the poor results of anti-PD-1 or anti PD-L1 monotherapy remains an unanswered question [18].

Higher ORRs were observed in patients treated in first line, especially if the tumor expressed PD-L1 and if a higher level of TIL infiltration was observed [19]. In the various cohorts of the Keynote 086 study (phase II), pembrolizumab monotherapy made it possible to obtain higher response rates in patients with PD-L1 positive tumors, in the first line of treatment (ORR: 23%, with 4% achieving complete response), compared to patients who had already received chemotherapy, and not selected for tumor PD-L1 status (ORR: 4.7%) [20]. These first results, suggesting a greater benefit during early use, and when the tumor expresses PD-L1 by immunohistochemistry, were confirmed in the phase III Keynote 119 study, in which pembrolizumab monotherapy was used in second or third line of metastatic TNBC treatment, without better ORR as compared to single-agent chemotherapy. A significant increase in survival was nevertheless seen for patients with higher tumoral PD-L1 expression [21]. Beyond mTNBC, avelumab, an anti-PD-L1, has been used as monotherapy in patients with different subtypes of metastatic breast cancer, and yielded an ORR of 2.8% for ER+/HER2− tumors, but no response in HER2 positive tumors (compared to an ORR of 5.2% in patients with mTNBC) [22]. 

These first results, therefore, seem to indicate that a small group of patients (difficult to identify due to imperfect selection biomarkers) with metastatic breast cancer (and not only of the TNBC subtype), may benefit from therapeutic approaches by ICB (immune checkpoint blockade) monotherapy, but also that it is necessary to develop therapeutic strategies in order to increase the number of responding patients. Thus, concerted efforts have been made to better understand the mechanisms responsible for defects in spontaneous immune response against BC, and how to make it more efficient from a therapeutic point of view, in order to rationally combine ICB with other treatments. 

Many treatments used in oncology are indeed capable of inducing immunogenic cell death (ICD) [23,24]. ICD is a mode of drug-induced cell death characterized by molecular hallmarks including the release of damage-associated molecular patterns (DAMPs) by dying cancer cells [25]. ICD can initiate efficient antitumor immunity, and thus augment the therapeutic effect of a given cytotoxic drug [26]. Besides this effect, some commonly used chemotherapies also appear able to modulate the tumor immune microenvironment quantitatively or qualitatively, influence antigen presentation or release of tumor antigens or chemokines, or are capable of remodeling the tumor stroma in a manner more favorable to immune cell trafficking [27]. Since many chemotherapy drugs are able to act at these different levels, and chemotherapy is part of the usual treatments for localized or metastatic breast cancer, numerous studies involving ICB+ chemotherapy have been performed, in TNBC and other subtypes.

Initial phase I-II trials testing combinations involved chemotherapies usually used for the treatment of metastatic breast cancer (MBC), in particular nab-paclitaxel [28], or eribulin mesylate [29], and clearly showed higher response rates (26–39%) than monotherapy with ICB, whether in the first or second line, again with lasting tumor response. Following these encouraging results, several randomized phase III trials were performed to confirm the value of adding anti-PD(L)-1 treatment to first-line chemotherapy in metastatic TNBC (mTNBC). However, currently available results from these studies are conflicting: the IMpassion 130 trial (NCT02425891), which randomized patients to nab-paclitaxel vs nab-paclitaxel + atezolizumab as first-line treatment for mTNBC, showed a significant benefit in terms of PFS in the overall population, but also an especially marked benefit in patients with a PD-L1 positive tumors (7.5 vs. 5.0 months, *p* < 0.001) [30]. A benefit in OS was not shown in the overall population, but an exploratory analysis in the PD-L1 positive population highlighted a median OS of 25 months in the atezolizumab group, compared to 15.5 months in the placebo group. These results contrast with those from the IMpassion 131 trial (NCT03125902), in which mTNBC patients were randomized between paclitaxel and paclitaxel + atezolizumab, with no advantage in terms of OS or PFS for patients receiving anti PD-L1 [31]. These results are particularly intriguing, and remain unexplained, especially since a third study (Keynote-355, NCT02819518) conducted in the same type of patient (first-line treatment of mTNBC) reported a clinically meaningful improvement in PFS among patients treated with pembrolizumab in combination with chemotherapy (nab-paclitaxel, paclitaxel or gemcitabine-carboplatin) [32], compared to chemotherapy alone. The pembrolizumab benefit was seen especially in patients whose tumor expressed significant levels of PD-L1 by IHC (CPS score ≥ 10) (median PFS 9.7 vs. 5.6 months). Interestingly, in this study, patients treated with paclitaxel + pembrolizumab seemed to benefit as much from ICB as those who received nab-paclitaxel + pembrolizumab, compared to patients treated with the same chemotherapies plus placebo. The reasons for these discordant results between phase III trials are not fully understood, and the results of other first-line studies incorporating ICB in association with standard of care are eagerly awaited (e.g., the IMpassion 132 trial NCT03371017 in mTNBC, or NCT03199885 in HER2+ BC). In this review, we will discuss some hypotheses that may explain these differences from an immunological point of view.

Taking into account immunosuppressive load, and tumor clonal heterogeneity, probably at a lower level as compared to metastatic disease, it would be expected to obtain deeper tumor responses in early breast cancer (eBC) with chemotherapy + ICB. As the efficacy of neoadjuvant chemotherapy (NAC) can be evaluated by pathologic complete response (pCR) in eBC, many clinical trials have evaluated the combination of chemotherapy + ICB in this setting. Earliest phase Ib studies, like Keynote 173 [33] showed impressive pCR rates of around 60% in PD-L1 unselected eTNBC patients. Higher pretreatment expression of PD-L1, and stromal TIL levels were significantly associated with higher pCR rates. I-SPY 2 is a phase II trial platform that is evaluating novel neoadjuvant agents on top of a classical NAC backbone of taxane/anthracycline, with pCR estimation by Bayesian models. With this approach, combining pembrolizumab with NAC made it possible to triple pCR probability not only in eTNBC (60% vs. 20%), but also in ER+/HER2- eBC (34% vs. 13%) [34]. These encouraging results have prompted larger randomized trials comparing NAC to NAC + ICB in eBC. In the phase II GeparNuevo trial [35], surprisingly, there was no difference in pCR between patients treated with NAC alone and NAC + durvalumab, except in those with a short run-in of durvalumab monotherapy before NAC + ICB (pCR: 61% vs. 44.2% in the placebo arm). These results highlight the need to better understand how to sequence ICB with chemotherapy, and to discover the optimal “immunologic window” in which to use immunotherapy during the treatment course. Of note, after extended follow-up, patients treated with durvalumab had better 3-year survival (distant-DFS, and OS) [36]. More recently, 2 phase III randomized trials clearly indicated a significant benefit of adding anti-PD-(L)1 to NAC for the treatment of eTNBC: in the IMpassion 031 trial, an absolute pCR increase of 17% was observed in patients randomized to the atezolizumab + NAC arm, compared to NAC alone (58% vs. 41%) [37]. In the Keynote 522 trial, patients treated with NAC + pembrolizumab had a pCR rate of 65%, significantly higher than in patients who received NAC only (51%) [38]. Interestingly, in these two positive studies, PD-L1 tumor expression was not predictive of ICB benefit, but was associated with a higher probability of achieving pCR, both with ICB + NAC, and with NAC alone. In contrast, in a third study called neoTRIP [39], no difference in pCR was observed between patients treated with NAC alone, and those treated with NAC + atezolizumab. An explanation for these apparently contradictory results could be the nature of the chemotherapy backbone, which comprised anthracyclines (sequenced with taxane +/− carboplatin) in IMpassion 031 and Keynote 522, but not in neoTRIP (paclitaxel + carboplatin). Given the differences in immunological properties of these drugs, which we will discuss in this review, these clinical results highlight the importance of considering the therapeutic partners of ICB from an immunological perspective. In eTNBC, many other large clinical trials evaluating PD-(L)1 blockade are ongoing, in the neoadjuvant setting (like NSABP B-59 NCT03281954), the adjuvant setting (Impassion 030 NCT03498716), or in post-neoadjuvant high risk patients with post-NAC residual disease (SWOG S1418/BR006 (NCT02954874), A-Brave (NCT02926196). The same neoadjuvant strategy incorporating pembrolizumab with standard NAC is also being explored in high risk ER+/HER2- eBC patients in the Keynote 756 study (NCT03725059).

Several lessons can be learned from these clinical and biological results. First, concerning the predictive value of PD-L1 status, all these studies show that it is not the perfect biomarker for predicting the benefit of immunotherapy. Moreover, while increased expression of PD-L1 makes it possible to select a population who probably benefit more from immunotherapy (in monotherapy or in combination with chemotherapy compared to chemotherapy alone), on the other hand in early stage disease (neoadjuvant setting), PD-L1 expression is probably linked to intra-tumoral immune response and TIL infiltration, which is a prognostic and predictive factor for chemotherapy efficacy. In TNBC, the differences in the efficacy of chemotherapy + immunotherapy combinations between localized and metastatic stages are probably partially explained by the immunosuppressive load, and increasing tumor heterogeneity with more advanced disease stage, [40,41,42]. Moreover, metastatic spread could also reflect escape from immune surveillance at the primary tumor site, and thus, the presence of immunoselected disease that may be less sensitive to immunotherapeutic approaches [43]. As “hot” immunogenic BC components are usually more chemo-sensitive, the metastatic disease is probably also enriched in “cold” tumor clones, previously selected by initial chemotherapy. For metastatic patients, the predictive value of PD-L1 expression is also partly related to the presence of an anti-tumor immune response, and suggests that it is essential to use therapeutic partners capable of converting a cold and/or immunosuppressed tumor microenvironment into a hot, activated immune context, in order to sensitize the tumor to PD-1 blockade. However, considering the immunological properties of paclitaxel (the most widely used first-line treatment in mTNBC), it is unlikely that this drug class is the best companion for immunotherapy in BC. Indeed, preclinical and human data have shown that paclitaxel is not a strong ICD inducer and has a limited effect on selective depletion/inhibition of immunosuppressive cells [27,44]. In the absence of a clear and substantial benefit of chemo-immunotherapy in metastatic BC, it seems essential to consider new therapeutic combinations by exploring the immune effects of different therapies. This “pick the winner” approach was recently tested in the TONIC trial [45], and showed that not all chemotherapies used in MBC have the same effect on the induction of an immune response amenable to enhancing the efficacy of immunotherapy with nivolumab. 

Several important points should thus be taken into account: (i) the immunological effect of the therapies conventionally used in BC (targeted therapies, chemotherapy, kinase inhibitors, etc.) and (ii) the immune tumor microenvironment. The immunological effects of drugs can vary widely, but the tumor itself can also modulate its environment in various ways, in quantitative, qualitative and topographical terms, to make it hostile to the immune response. This leads to the appearance of different tumor subgroups, namely inflammatory tumors termed “hot” tumors; non-inflammatory or “cold” tumors, and “immune-excluded” tumors. A hot tumor is described as a tumor highly infiltrated by immune cells and frequently associated with high PD-L1 and interferon-γ (IFNγ) signature expression (and therefore, a priori, sensitive to anti-tumor immunity). Conversely, a cold tumor is characterized by a paucity, or total lack of immune cell infiltration and anti-tumor immune response [46]. Immune-excluded tumors are tumors where the action of the immune system is limited by the presence of a physical barrier that prevents the immune cells from infiltrating the tumor. There is thus a strong rationale for combining immunotherapy comprising an anti-PD-L1/PD1 checkpoint inhibitor with therapies capable of converting a cold tumor into a hot tumor, enabling increased expression of PD-L1. On the other hand, it also seems possible to combine ICB with some conventional therapies capable of eliminating the physical barrier, thereby activated and efficient immune cells to infiltrate the tumor. 

The objective of this review is to take stock of preclinical and clinical data in order to identify, among the therapies used in breast cancer, those most relevant to sensitize: (i) non-immunogenic or non-antigenic cold tumors, (ii) immunosuppressed cold tumors, and (iii) immune-excluded tumors. There are currently a large number of clinical trials testing therapeutic combinations with immunotherapy in breast cancer, most of them evaluating combinations incorporating PD-(L)1 inhibitors [47].

## 2. Turning a Non-Antigenic and Non-Immunogenic “Cold” BC Tumor into a “Hot” Tumor

It now well recognized that the quantitative and qualitative composition of the tumor immune micro-environment influences not only the prognosis of the disease but also the effectiveness of certain treatments such as ICI. Patients with apparently the same type of tumor can in fact have immunologically very different immune TME, which is important to know in order to adapt the therapeutic strategy. In a simple way, tumors can be divided into two main categories according to their immune TME. “Hot” tumors are characterized by a strong infiltration of T lymphocytes, transcriptomic signatures of a favorable inflammatory response (interferon-γ), and frequently a strong expression of PD-L1 testifying to this immune response. These tumors have a high probability of responding well to therapeutic approaches based on ICI. Conversely, cold tumors are characterized by low T lymphocyte infiltration (immune desert) or T infiltration remaining at the periphery of the tumor (immuno-excluded phenotype), associated with the absence of other signs of an effective inflammatory response [48,49]. Several mechanisms can explain the “cold” tumor phenotype associated with resistance to PD-(L)1 blockade. The two main ones are: (i) loss of antigenicity, and/or (ii) loss of immunogenicity. 

Poorly antigenic tumor can be explained by a low tumor mutational burden (TMB), which is common in BC [50], and associated with a lower probability of expressing tumor-specific antigens (neoantigens). Tumor cells can also lose their capacity for antigen processing and/or presentation, for example after genetic deletion of MHC-I loci or by *JAK1/2* mutation and subsequent inhibition of INFγ-induced MHC-I expression [51,52,53,54]. Reduced expression of MHC I has been observed in human breast cancer samples [55]. 

The absence of immunogenicity may be due to low release during ICD of DAMPs such as extracellular high mobility group box 1 (HMGB1), adenosine tri phosphate (ATP), calreticulin (CRT), or double-stranded DNA (dsDNA) (Figure 1), and/or the induction of inhibitory pathways, such as toll like receptor 4 (TLR4) signaling inhibition, CD39/CD73, CD47 or Trex expression [56]. Accordingly, tumors with both poor antigenicity and a low level of inflammation are generally much more resistant to immunotherapy [57]. This is the case of the majority of BC, which present both a low TBM and a low level of inflammation, unlike immunogenic NSCLC or melanoma tumors [58]. Therefore, identifying therapies that can upregulate antigenicity and/or immunogenicity in breast tumors is essential to sensitize these tumors to immunotherapy, particularly inhibitors of the PD1/PD-L1 axis. The first part of this review will therefore focus on the effects of existing therapies capable of stimulating these two biological processes.

### 2.1. Therapeutic Approaches to Increasing Antigenicity

#### 2.1.1. Poly ADP Ribose Polymerase Inhibitors (PARPi)

PARPi (olaparib, talazoparib) are currently approved for MBC patients with deleterious germinal BRCA1/2 mutations. These tumors with homologous repair deficiency and treated with PARPi are characterized by high genomic instability, and high numbers of DNA breaks, thus generating dsDNA fragments able to activate the interferon pathway (and thus increased MHC expression) in treated cancer cells by stimulating the STING (Stimulator of Interferon Genes) pathway [60,61]. Moreover, it has been postulated that this high genomic instability could also generate non synonymous mutations, translating into potentially immunogenic neoantigens. These BRCA-mutated tumors have been also described as being more infiltrated by immune cells [62].

Additionally, in cancer cell lines, xenograft and syngeneic mouse models of BC, administration of PARPi (olaparib and talazoparib) has been shown to induce PD-L1 expression [63]. In the context of BRCA2-deficient BC, Sato et al. reported that PARPi-induced dsDNA breaks could directly regulate PD-L1 through the ATM-ATR-Chk1 pathway, independently of the type I interferon pathway [64]. A further study found that upregulation of PD-L1 was mainly linked to the increase in anti-cancer immune response and relied on activation of the STING/TBK1/IRF3 pathway by cytosolic dsDNA generation after olaparib treatment [65]. Consequently, preclinical studies shown that the combination of PD-L1 blockade plus PARPi is additive against tumor growth in different TNBC and ovarian models [60,63]. Based on the encouraging results of these preclinical studies, a series of clinical trials are underway to assess the efficacy of PARPi in combination with ICI in a wide range of cancers, including BC. 

The association of durvalumab (an anti PD-L1) with olaparib in patients with germline BRCA1/2 metastatic ER+ or TNBC yielded a disease control rate of 80% at 12 weeks (primary efficacy endpoint), with a median duration of response of 9.2 months in the phase I/II MEDIOLA [66].

The phase II Keynote-162/TOPACIO study evaluated the effect of a combination of niraparib and pembrolizumab in mTNBC [67] (NCT02657889). This combination appeared to be safe and provided an interesting antitumor effect (29% overall response rate), particularly in patients with germline BRCA mutation (67% overall response rate) [67]. Moreover, PD-L1-positive tumors responded better than PD-L1 negative (33% vs. 15%). 

#### 2.1.2. MEK Inhibitors

Multiple signaling pathways involved in oncogenesis can be activated in parallel to the PD-L1 pathway during tumor immune escape. For example, the MAPkinase pathway is involved in the fight against inflammation by inhibiting the secretion of pro-inflammatory cytokines such as IFNγ [68]. An interesting opportunity for therapeutic synergy is, therefore, to target oncogenic pathways when these also participate in immune escape. Accordingly, inhibition of the MAP kinase pathway (frequently activated in many solid tumors including BC) using a MEK inhibitor seems capable of increasing expression of MHC I and II, PD-L1 expression, tumor infiltration by CD8^+^ T cells, thereby sensitizing tumors to anti-PD-1 immunotherapy [69,70].

It was recently observed that a combination of anti-PD-(L)1 and a MEK inhibitor (trametinib) yielded an increase in MHC class I and II expression, and of PD-L1 in vitro in human and murine mammary cancer cells, as well as in vivo in mice [71]. In addition, one study showed that, by adding trametinib to anti-PD1 treatment in mice, a significant reduction in tumor volume was observed [70].

Conversely, the recent phase II COLET study combining paclitaxel, atezolizumab and cobimetinib (a MEK inhibitor) did not show any additional efficacy compared to treatment with paclitaxel and atezolizumab only [72]. A possible explanation for these disappointing results is that MEK inhibition could be detrimental to the functionality and activation of T lymphocytes [73]. These preclinical data illustrate the difficulty of targeting a cellular pathway that may have opposing roles in immune and tumor cells. Other BC studies are under way with various MEK inhibitors such as selumetinib, which showed an acceptable toxicity profile in combination with an mTORC1/2 inhibitor (vistusertib) in a phase Ib/IIa trial, with lasting stability in mTNBC patients [74]. A phase I/II clinical trial of pembrolizumab plus binimetinib is currently ongoing in the setting of local and metastatic triple negative breast cancer (NCT03106415).

#### 2.1.3. CDK4/6 Inhibitors

CDK4/6 inhibitors such as palbociclib, ribociclib, and abemaciclib have led to improved progression-free and overall survival in HR+ patients [75]. Some studies show that treatment with CDK4/6 inhibitors increases MHC class I expression by BC cells. CDK4/6 inhibitors work primarily by suppressing the phosphorylation of retinoblastoma protein (Rb) in cancer cells, which stops the cell cycle and inhibits proliferation. Besides these anti-proliferative effects, it has been shown that CDK4/6 inhibitors can increase antigen presentation by MHC class I molecules in models of BC lines, and increase expression of MHC class I and II molecules, partly through re-expression of endogenous retroviral sequences. CDK4/6 inhibitors are also able to induce cell-cycle arrest and tumor cell senescence, thus leading to the activation of the SASP (senescence-associated secretory phenotype); which in turn can induce recruitment of immune cells into the tumor microenvironment [76,77,78].

In these mouse models, CDK4/6 inhibitors synergized with immunotherapy, or immunogenic chemotherapy and increased survival in treated animals. The first phase I/II clinical study to show favorable results tested abemaciclib and pembrolizumab in patients with HR + metastatic BC, and reported four patients (14%) with objective response at 24 weeks [79]. Similar studies are under way in other types of cancer.

#### 2.1.4. Combinations with Other Immunotherapies

Targeting the PD-1/PDL1 axis as the sole immunotherapy approach is arguably insufficient to reinvigorate the anti-tumor immune response in many patients.

Complementary immunotherapy approaches to anti-PD-1/PD-L1 must therefore be devised in order to increase the presentation of tumor antigens, increase tumor infiltration into immune cells, and/or increase the activity of cellular effectors.

These alternative immunotherapy approaches make use of anti-tumor vaccines, cytokines, or molecules targeting other activating or inhibiting checkpoints of the immune response. The rationale for their combination with anti-PD-1/PD-L1 has been the subject of many recent reviews, e.g., [80].

To date, relatively few combined immunotherapy approaches have been reported in BC. However, many studies have been launched (with T-cell targeted immunomodulators, other immunomodulators, cancer vaccines, oncolytic viruses, or T-cell targeted bispecific mAb) [47]. Antibodies targeting LAG3 are one example of the type of immunotherapy aimed at increasing antigenicity: LAG3 is a surface molecule that binds to MHC class II on antigen presenting cells, thereby preventing T cells from binding to MHC class II, thus preventing their activation. A number of antibodies targeting LAG3 are under development, as well as bispecific antibodies that engage both LAG-3 and PD-1/L1 (NCT03219268, NCT03440437) (Table 1).

In addition, antibodies targeting PD-(L)1 checkpoint inhibitors can also be combined with an antibody targeting the CTLA-4 molecule. The CTLA-4 receptor functions as an immune checkpoint to moderate the immune response. It acts as a switch that will inhibit the action of the lymphocyte when it comes into contact with CD80 or CD86 proteins on the surface of an antigen presenting cell [81]. Thus, an anti-CTLA-4 strategy unblocks the antigenic priming phase.

Dual co-inhibition of anti-CTLA-4 and anti-PD-1/L1 has shown improved PFS and OS in melanoma, and there is preclinical data to support its use in BC [82]. A pilot trial of durvalumab (an anti-PD-L1) and tremelimumab (an anti-CTLA-4) in metastatic BC observed an ORR of 17% in ER+ patients, but an ORR of 43% in patients TNBC, suggesting that patients with TNBC may be better candidates for this type of treatment [83]. Nevertheless, the toxicity of this double inhibition remains a concern, as a recent analysis of trials in melanoma/renal carcinoma showed increased efficacy but a near doubling of grade 3–4 toxicity compared to Ipilimumab as monotherapy. Dual bispecific immunomodulators combining two inhibitory functions are under study [84]. A phase I trial on the XmAb20717 molecule, a combined PD-1 and CTLA-4 antibody in certain advanced solid tumors is underway (NCT03517488).

The discovery of ICIs involved in T lymphocyte depletion such as LAG-3 or TIGIT has also enabled the development of treatments targeting these molecules, some of which are being tested in advanced BC, in particular TNBC (NCT03742349, NCT01968109, NCT04252768, NCT03971409, NCT02794571, NCT04584112).

#### 2.1.5. Targeting Tumor Antigenicity with Immunological Synergy: The Paradigm of HER-2 Directed Monoclonal Antibodies

The HER2 oncogene is amplified in about 15% of BCs, and this tumor associated antigen (TAA) constitutes a target for HER-2 directed monoclonal antibodies (mAbs). These mAbs, e.g., trastuzumab, constitute a pivotal axis of treatment for this BC subtype. In the past few years, it has become clear that part of the antitumor activity of trastuzumab is in fact mediated by the immune system, especially by antibody-dependent cellular cytotoxicity (ADCC). Several studies have shown that induction of ADCC is associated with an increase in TILs within the tumors of patients treated with trastuzumab [9,85,86]. Similar observations have been made in preclinical mouse models, and in patients with other HER2-targeting mAbs, such as pertuzumab and T-DM1 (an antibody-drug conjugate) [87,88]. In preclinical models of BC, these treatments synergize with diverse immunotherapies, including anti PD-(L)1 [89]. 

Clinical responses have been observed with the association of trastuzumab and pembrolizumab in patients harboring trastuzumab-resistant tumors [90]. The combination of T-DM1 + atezolizumab also seems feasible, without additional toxicity, but did not show any benefit compared to T-DM1 alone in the KATE-2 study, except in patients with a PD-L1 positive tumor, in whom there was a slight benefit in PFS [91].

These clinical results are ultimately quite disappointing, but trials are still recruiting patients at earlier stages of the disease, especially in the first line of metastatic setting (NCT03199885).

### 2.2. Therapeutic Approaches to Increasing Immunogenicity

After seeing the impact of different therapeutics on the antigenicity of tumor cells, a second step may be involved in the success of combination treatments containing immunotherapy, namely tumor cell immunogenicity. This concept has been demonstrated in mouse models of colorectal cancer treated with anthracyclines [92]. After treatment, and in addition to the purely cytotoxic effect, dying tumor cells may emit danger signals, characterized in particular by the extracellular release of HMGB1 or ATP, membrane calreticulin (CRT) exposure, or secretion of type I interferons and chemoattracting chemokines (CXCL10) (Figure 1). These “danger” signals are intended to alert and mobilize the immune system to recognize antigens and eliminate tumor cells. The association of an immunogenic therapy with immunotherapy therefore makes sense, by enabling the recruitment and activation of immune cells within the tumor [26].

#### 2.2.1. Chemotherapies

Among the chemotherapies currently used in BC, anthracyclines, notably doxorubicin, epirubicin or mitoxantrone, are molecules known to induce ICD in different tumor models [92,93]. ICD is considered as a stress response associated with the release of diverse DAMPs (CRT, HMGB1, ATP etc., Figure 1) in addition to enhanced antigen presentation [26,27]. Among the DAMPs, HMGB1 release and CRT re-localization has been established in BC following exposure of human tumor cells in vitro, but also in vivo in patients treated with doxorubicin or paclitaxel [94]. Moreover, the efficacy of anthracycline-based adjuvant chemotherapy has been shown to be associated with the integrity of the HMGB1/TLR4 axis [95,96]. In a trastuzumab-resistant HER2+ breast cancer model, an anti-HER2 conjugate carrying an anthracycline derivative was shown to induce ICD. Further, combining this with an anti-PD-1 antibody improved tumor regression after treatment [97]. The combination of NKT cell activation with chemotherapy, e.g., gemcitabine or cyclophosphamide (currently used in BC treatment), enhanced the immunogenicity of breast tumor cells by increasing ICD signals (CRT, ATP, HMGB1) in metastatic breast cancer [98]. 

A fourth hallmark of ICD has recently been discovered, linked to autocrine signaling of type I interferons (INF-I). The release of nucleic acids by anthracycline-killed tumor cells is detected by TLR3 on persistent viable cells, and acts as a transcription signal for type 1 interferon genes. IFN-I in turn acts on their IFN-I receptors in an autocrine manner, leading to an autocrine IFN-I signature that comprises essential chemotactic factors for the recruitment of immune effectors, like CXCL10 [99]. 

Expression of chemoattracting chemokines seems crucial for the efficacy of ICIs [100], and it is therefore important to know that certain chemotherapies, such as anthracyclines, are likely to induce their expression.

However, although chemotherapy and anti-PD1 synergy has been shown in many preclinical models, few clinical trials have compared different chemotherapy regimens in association with the same immunotherapy.

In the phase III Keynote 355 trial, the addition of pembrolizumab to chemotherapy as the first treatment for patients with mTNBC showed an improvement in PFS regardless of the associated chemotherapy (gemcitabine + carboplatin or different taxanes), but there was no arm with anthracyclines [32]. Conversely, at advanced stages of the disease, doxorubicin and cisplatin were shown to be more likely to give synergic responses in combination with nivolumab in the phase II TONIC trial [45].

Interestingly, the only negative trial (no improvement in pCR with immunotherapy) combining chemotherapy + immunotherapy in the neoadjuvant treatment of TNBC, is the one in which neoadjuvant chemotherapy did not include anthracyclines [39].

There is therefore a growing body of indirect evidence suggesting that chemotherapeutic agents capable of inducing ICD can synergize more effectively with anti PD-(L)1.

#### 2.2.2. Radiotherapy

Radiation therapy is another type of treatment used in BC. In addition to inducing lethal DNA damage in tumor cells, it appears to have an effect on tumor immunity. Indeed, the exposure of tumor cells in vitro and in vivo to different doses of radiation appears to lead to an increase in ICD signals and antigen processing machinery [23,100]. Radiotherapy is able to induce ICD of tumor cells, in connection with danger signals emitted as a result of DNA damage caused by radiation [101]. This could explain the abscopal effect of radiotherapy, observed in certain patients [102]. Moreover, in preclinical models of TNBC, Dovedi et al. showed that low dose fractionated radiotherapy could upregulate PD-L1 ligand expression by tumor cells [103]. The combination of radiotherapy and anti-PD-(L)1 immunotherapy produced effective anti-tumor immunity and long-term control of the tumor. BC is a disease that is usually radiosensitive, and radiotherapy is part of the standard treatment for localized breast cancer. Therefore, the association of radiotherapy with immunotherapy could be synergic in BC, as has been shown in lung cancer, for example, in the PACIFIC phase III trial [104]. Many association studies including radiotherapy are currently under way, and have recently been reviewed [105]. In BC, the recently published results of a phase II trial show an encouraging response rate (17.6%) in heavily pretreated patients when pembrolizumab is added to palliative irradiation of a metastatic site [106,107].

Importantly, it should be noted that the optimal radiation therapy treatment plan (dose and fractionation) to obtain optimal immunological synergy is currently unknown.

Depending on the scheme used, radiotherapy may also have immunosuppressive effects [108], part of which can be counteracted by anti-PD-1 drugs [102,103,109]. The timing of the radiation therapy in relation to the administration of immunotherapy (concurrent or sequential), and the administration schedule of the radiotherapy are possibly at the origin of variable immunological consequences [23,103,108,109] and must be taken into account in the therapeutic combinations. Research is currently ongoing to try to answer these new questions, and to investigate the integration of stereotactic radiotherapy into combination strategies with immunotherapy [110].

#### 2.2.3. STING Agonists

STING agonists have emerged as good candidates to promote the recruitment of immune cells in the tumor microenvironment. Indeed, the activation of this pathway leads to the production of type I interferons and an adaptive immune response [111]. Injection of cGAS-NP, a liposomal STING inducer, in preclinical BC models resistant to anti-PD-L1 immunotherapy showed a strong increase in IFNβ expression and secretion, as well as STING/interferon α/β receptor (IFNAR)-pathway-dependent tumor regression [112]. The generation of an adaptive immune response was demonstrated by the rejection observed following re-challenge of tumor cells from different mouse models, such as the 4T1 TNBC, proving that administration of STING agonists provided long-lasting immune memory [113]. The combination with anti-PDL1 immunotherapy is promising because, as proven in melanoma, cGAMP would activate and recruit CD8^+^ T cells in the microenvironment, and cGAS seems essential for the success of PD-L1 treatment [114]. Interestingly, there seems to be immunological synergy between these treatments, and others capable of strongly inducing breaks in DNA, such as PARPi (especially in BRCA-associated BC) [115].

Currently, there are few available human clinical data in BC and no clinical trials are currently open.

## 3. Targeting the Immunosuppressive Microenvironment

In addition to cancer cells, tumors contain a repertoire of recruited immune and non-immune cells that contribute to the creation of the “tumor microenvironment”. Among these populations, certain cell types have immunosuppressive action, such as macrophages associated with type 2 tumors (TAM2), regulatory T lymphocytes (Treg), tumor associated neutrophils (TAN), Myeloid-Derived Suppressor Cells (MDSC), or cancer associated fibroblasts (CAF). Certain therapies are capable of inhibiting or depleting these immunosuppressive populations, but also of inhibiting the cytokines produced by these immunosuppressive cells. Targeting these cell populations could help to reduce immunosuppression within the tumor, which would in turn sensitize the tumor to immunotherapy via checkpoint inhibitors.

### 3.1. Chemotherapies

Besides the possible induction of ICD, some chemotherapies are known for their depleting effects on immunosuppressive populations, such as cyclophosphamide for Treg lymphocytes [116] or gemcitabine or 5-fluorouracil for MDSC [117]. A preclinical evaluation of a combination of anti-PD-L1 and cyclophosphamide in a mouse model of BC failed to demonstrate a superior effect over cyclophosphamide monotherapy [118]. In contrast, the phase II CHEMOIMMUNE trial of pembrolizumab following treatment with metronomic cyclophosphamide is currently underway in lympho-penic patients with metastatic BC (NCT03139851). Other chemotherapies are able to modify the phenotype of immunosuppressive cells. Indeed, paclitaxel enables repolarization of M2 type macrophages into the M1 antitumor phenotype in a TLR4-dependent manner; this has been shown both in vitro and in vivo in preclinical studies in a 4T1 BC model [119]. In addition, another study revealed that paclitaxel at ultra-low concentrations enables the differentiation of MDSCs into dendritic cells in vitro in a TLR4-independent manner [120]. 

### 3.2. IDO Inhibitors

Indoleamine 2,3-dioxygenase (IDO) is an enzyme that converts tryptophan to kynurenine, exerting an immunosuppressive effect within the tumor microenvironment. Indeed, kynurenine is an enzyme capable of inactivating effector T lymphocytes and promoting regulatory T lymphocytes [121]. Like PD-L1, IDO is upregulated by T cells secreting IFNγ in the microenvironment as a means of immune escape, and both of these pathways are potentially redundant pathways of immune suppression in BC that present TILs. The combination of indoximod, an orally administered IDO inhibitor, and docetaxel has been tested in solid tumors including BC. The results showed evidence of clinical activity: two partial responses and two minor responses in BC were observed [122] (NCT01792050). Several clinical trials are evaluating the activity of indoximod in combination with anti-PD1 in multiple types of tumors, including BC. Epacadostat is another inhibitor of IDO that recently showed a response rate of 10% in combination with pembrolizumab in mTNBC [123].

### 3.3. The CD39/CD73/Adenosine Pathway

In BC, adenosine, produced by the CD39 and CD73 ectonucleotidases, mediates numerous molecular pathways of immunosuppression, especially inhibition of T cell proliferation, cytotoxicity, or cytokine production [124,125]. Ectonucleotidase CD73 is particularly expressed in TNBC, and is associated with chemo-resistance [126].

In a preclinical model of TNBC, inhibition of adenosine receptors synergizes with anti-PD-1 in a CD8^+^ T cell/NK cell/ and interferon-dependent manner [127]. Oleclumab, an inhibitor of CD73, is currently under clinical evaluation in combination with paclitaxel + carboplatin + durvalumab (anti PD-L1) in the phase I/II SYNERGY in mTNBC patients (NCT03616886).

### 3.4. PI3K/AKT Inhibition

The PI3K/mTOR/AKT pathway is one of the most frequently mutated pathways in BC, prompting the development of various pharmacological inhibitors (PI3 kinase, AKT or mTOR) for BC treatment. Moreover, this pathway also participates in the development of an immunosuppressive environment by increasing the expression of immunosuppressive cytokines and chemokines. This occurs via the recruitment of MDSCs and regulatory T lymphocytes within the tumor, and through an increase in PD-L1 ligand expression by tumor cells [128,129]. Inhibitors targeting this pathway have been developed in recent years such as ipatasertib (an AKT inhibitor), whose first results show a decrease in the number of regulatory T lymphocytes within the tumor accompanied by an increase in differentiation of CD8^+^ lymphocytes into CD8^+^ memory [130]. In addition, it has been shown that PI3K inhibitors enabled polarization of M2 macrophages from an immunosuppressive phenotype into M1-type macrophages of the antitumor phenotype. Inhibition of PI3Kγ appears to sensitize tumors to anti-PD-1 antibody therapy and to slow tumor growth by increasing the level of CD4^+^ and CD8^+^ T lymphocytes within the tumor [131]. These encouraging preclinical results have prompted the development of these associations in the clinic. Indeed, a study testing ipatasertib in combination with atezolizumab and paclitaxel in patients with localized or metastatic TNBC is currently underway (NCT04177108). Another phase II clinical study combining a PI3K-γ inhibitor, IPI-549 (because of the important role of this PI3K isoform in the functions of immunosuppressive myeloid cells), with atezolizumab and nab-paclitaxel is ongoing (NCT03961698).

### 3.5. HDAC Inhibitors

A number of pharmacological agents capable of modifying epigenetics, such as HDAC inhibitors, are currently being investigated in BC. 

Besides their direct effect on tumor cells (apoptosis, cell differentiation, growth inhibition), there is great interest in the potential of epigenetic therapy to prime the response to immunotherapy in BC. Studies have shown that epigenetic modulation can promote an IFN type I response and restore production of Th1-type cytokines and chemokines [132,133]. Another preclinical study showed that treating mice bearing solid tumors (including tumors generated with the 4T1 mouse model) with the HDAC inhibitor entinostat, combined with CTLA-4 and PD-1 antibodies, could eradicate both primary tumors and metastases by reducing granulocytic MDSCs [134]. HDAC inhibitors are thus able to reduce the immunosuppressive activity of Tregs and MDSCs [134,135].

A phase II clinical trial showed that adding entinostat to exemestane in patients with advanced ER+ BC resulted in an 8.3-month improvement in OS compared to patients treated with exemestane alone [136]. Exploratory studies on blood samples from 34 patients showed a lower number of MDSCs, a decrease in CD40 expression of MDSCs, and an increase in MHC class II expression on CD14^+^ monocytes two weeks after treatment initiation. No alteration of T cell phenotypes was observed. Multiple clinical trials are under way evaluating the combination of epigenetic modulation with PD-(L)1 blockade, or combined blockade of CTLA-4 and PD-1 [137].

### 3.6. CDK4/6 Inhibitors

Beside their effects on tumor antigenicity, CDK4/6 inhibitors increase the function of effector T lymphocytes while markedly suppressing the proliferation of regulatory T lymphocytes [138]. Preclinical and clinical studies have confirmed the increased tumor infiltration by T cells [139] and the decrease in Tregs in treated tumors [140].

### 3.7. Autophagy Inhibition

Certain immunosuppressive populations and their impact on the tumor microenvironment, such as angiogenesis or hypoxia, can promote cancer stem cell (CSCs) resistance [141]. Conversely, CSCs can also interact with these immunosuppressive populations to activate and stimulate them, such as inducing the recruitment and polarization of TAM2 [142]. CSCs are known to be involved in the metastatic capacity of cancers, particularly in breast cancer [143]. Indeed, through their capacity for self-renewal, differentiation and proliferation, these cells will promote the aggressiveness of cancerous lesions. One strategy to target CSCs is the inhibition of autophagy. Indeed, autophagy appears to promote the maintenance and resistance of CSCs in tumors to anti-cancer therapies [144,145]. Targeting this catabolic process could therefore become an interesting approach to limit resistant CSCs. Numerous phase I/II clinical trials evaluating the efficacy of autophagy inhibitors such as chloroquine or hydroxychloroquine are currently being investigated in the treatment of breast cancer [146]. Autophagy inhibition has also shown a positive impact on inflammation and recruitment of cytotoxic populations and appears to improve the efficacy of immunotherapy and justify its combination [147,148].

## 4. Counteract Immune-Excluded Tumors

### 4.1. Fibrosis

Some tumors manage to escape the immune system thanks to the formation of a barrier around their perimeter that prevents the arrival of immune cells within the tumor bed. The formation of this barrier can be explained by the marked fibroblast activation by tumor cells and other immunosuppressive cells, via the influence of TGF-β1. Cancer-associated fibroblasts (CAF) and tumor cells produce collagen and matrix proteins that participate in the formation of a fibrous capsule, also called peritumoral fibrosis. Interestingly, in TNBC, specific tumor microenvironment profiles (including a margin-restricted profile of CD8 infiltration) are linked to different transcriptomic subtypes of TNBC [149]. A margin-restricted profile of immune cells is more frequently observed in the mesenchymal stem-like TNBC subtype. This fibrotic barrier consisting of dense tissue enables the tumor to evade the immune system [150]. Furthermore, TGF-β1 produced within the tumor microenvironment has a central role in immunosuppression, as this cytokine can alter the efficacy of anti-tumor populations and promote pro-tumor populations [151]. Targeting this molecule could therefore make it possible to sensitize BCs to other immunotherapeutic approaches, including anti PD-(L)1. Several methods are currently available to target TGF-β1, such as monoclonal antibodies or TGF-β1 receptor inhibitors. Tauriello et al. have shown that in mouse models of colon and liver cancer, treatment with galunisertib, a TGF-β1 receptor II inhibitor, resulted in an increase in T-CD8^+^ cell activation, associated with a decrease in the number of metastases. In contrast, there appears to be no improvement in survival when targeting TGF-β1 alone [152]. In addition, immune system involvement, specifically CD8^+^ T cells, was found to be involved in the response to galunisertib treatment in a 4T1 mouse model of BC [153]. Bhola et al. combined galunisertib with paclitaxel treatment in mice, and observed a decrease in tumor volume with inactivation of the SMAD pathway, compared to paclitaxel treatment alone [154]. In BC, a combination of galunisertib and paclitaxel is currently being tested in patients with TNBC (NCT02672475).

Targeting TGF-β1 is also possible with the use of monoclonal antibodies. Preclinical use of dual immunotherapy comprising an anti-TGF-β1 and an anti-PD-L1 significantly increases mouse survival [155].The authors proposed that TGF-β1, by repressing lymphocyte infiltration within the tumor, prevents the action of anti-PD-1 or anti-PD-L1 treatment [155]. In humans, an anti-TGF-β1 antibody, fresolimumab is being tested in clinical trials, particularly in combination with radiotherapy in MBC. Results to date showed that patients receiving the highest dose of fresolimumab had a favorable systemic immune response and longer median OS than the lowest dose group (NCT01401062) [156].

In parallel, M7824 (bintrafusp alfa), a bifunctional fusion protein targeting TGF-β and PD-L1, is currently being tested in phase II and III clinical trials for many types of cancer. Compared to dual immunotherapy, M7824 showed greater anti-tumor efficacy in the EMT6 preclinical model [157]. Clinical studies are evaluating the efficacy of M7824 in HER2+ breast cancer (NCT03620201) and in TNBC (NCT04489940). In addition, the combination of this bispecific antibody with other therapies such as chemotherapy and radiotherapy is also being investigated. The effects of the combination of M7824 and chemotherapy with eribulin are currently being studied in TNBC (NCT03579472M). Lastly, the combination of radiotherapy and the M7824 antibody is currently being investigated in metastatic ER+/HER2− BC patients (NCT03524170).

### 4.2. Tumor Angiogenesis

Angiogenesis is the process of creating new vessels. During tumor growth, angiogenesis is necessary to reduce hypoxia and increase nourishment for tumor cell development [158]. Vascular endothelial growth factor (VEGF) and fibroblast growth factor (FGF) are two angiogenic growth factors involved in tumor progression [159]. In addition to reducing the density of neo tumor vessels, leading to hypoxia in cancer cells, anti-angiogenic therapies could help to normalize the tumor vasculature, thus facilitating immune cell trafficking in the tumor [160]. Moreover, certain agents targeting VEGF or its receptors have also shown that they could selectively deplete certain populations of immunoregulatory cells, as well as increase tumor expression of PD-L1 [161].

Pre-clinical data suggest that targeting the proangiogenic cytokines Vascular endothelial growth factor A (VEGF-A) and angiopoietin-2 (Ang-2) by a bispecific blocking antibody in an inducible model of BC leads to an increase in PD-L1 expression on epithelial cells in response to the induction of IFNγ and cytotoxic T lymphocytes. In this setting, combining this approach with anti-PD-L1 immunotherapy yields a prolongation of survival in 30% of mice compared to monotherapy [162]. 

The MORPHEUS phase 1b/II clinical trial is currently recruiting patients HR+HER2− or TNBC to evaluate the efficacy and toxicity of combining treatments with immunotherapy, such as atezolizumab + bevacizumab + endocrine therapy or atezolizumab + bevacizumab + anti-CD40 (NCT03280563) (NCT03424005). The ECLIPSE II clinical trial is evaluating the effects of combination immunotherapy in ER^+^HER2^−^ patients. Atezolizumab is combined with anti-MEK, anti-PI3K or bevacizumab+anti-MEK (NCT03395899). Another phase II clinical trial was conducted in women with TNBC treated by an anti-PD1 in combination with a VEGFR2 inhibitor. This combination seems to be effective, and to display strong synergy, with an ORR of 43.3% vs. <20% for monotherapies (NCT03394287) [163].

New anti-angiogenic compounds like famitinib are also in development in association with a new anti-PD-1, camrelizumab, in mTNBC with nab-paclitaxel as the chemotherapy backbone (NCT04129996).

## 5. Other Approaches

Innovative new approaches to immunotherapy are being developed with the aim of “heating up” cold or immune-excluded tumors. Among the options being investigated is intra-tumor administration of talimogene laherparepvec (T-VEC), an oncolytic herpes simplex 1 virus approved for the treatment of melanoma. A phase 1 trial of T-VEC in combination with neoadjuvant chemotherapy for the treatment of non-metastatic TNBC recently reported that this combination was feasible and led to a complete response rate of 55% [122]. Other trials associating T-VEC with ICI are ongoing, for example TVEC + atezolizumab in residual disease after standard neoadjuvant chemotherapy (NCT03802604).

Adoptive cell therapy using CAR-T cells or TILs in association with anti PD-(L)1 ICI are starting to be developed in BC, and some case reports have shown that this strategy can give strong and lasting tumor responses in very selected patients [164]. In mTNBC, numerous phase 1 trials are currently ongoing with CAR-T cells directed against various tumor antigens (mesothelin, cMET, MUC1, …) [165].

It should be noted that many new effective cytotoxic treatments are currently being developed in BC in the form of antibiotic-drug conjugates (ADC): even in the absence of a strong immunological rationale, several combination trials are already testing anti-PD-1 with sacituzumab govitecan (an antibody targeting the human trophoblast cell-surface antigen 2) (NCT03424005, NCT04468061), or ladiratuzumab vedotin [166].

## 6. Conclusions

Clinical results with anti-PD-(L)1 immunotherapy in breast cancer are mixed. It seems that monotherapy with these ICIs is clearly not sufficient to induce a deep and lasting response in the majority of patients, making it necessary to combine them with other treatments to achieve therapeutic synergy (Figure 2).

A large number of association trials are currently under way in all BC subtypes, and, in parallel, in metastatic and in early stage disease. However, it is worrying to note that all these efforts are moving forward in an uncoordinated manner, and most often without a strong immunological rationale.

It is quite probable that treatments based on immunotherapy only work in certain subgroups of BC subtypes and/or patients. Consequently, identifying these different clinical-biological entities, and their respective probability of response, should be a priority, in order to select the patients who may most benefit now from existing drugs (to clearly confirm the benefit), and conversely to prioritize access to clinical research programs for others, with a view to increasing the probability of response through therapeutic associations. In addition, the optimal schedule of administration of ICIs targeting PD-1 or PD-L1 with other therapeutic partners, their respective doses, and the treatment sequence are points that remain to be clarified.

The challenges to be faced in the years to come are daunting if immunotherapy is to reach its full potential in breast cancer. We must continue our search to find the best possible combination for a given patient, depending on the subtype of disease, the tumor immune microenvironment, and the patient’s immunological context (degree of immunosuppression, microbiota, comedications, etc.).

## Figures and Tables

**Figure 1 cancers-13-05999-f001:**
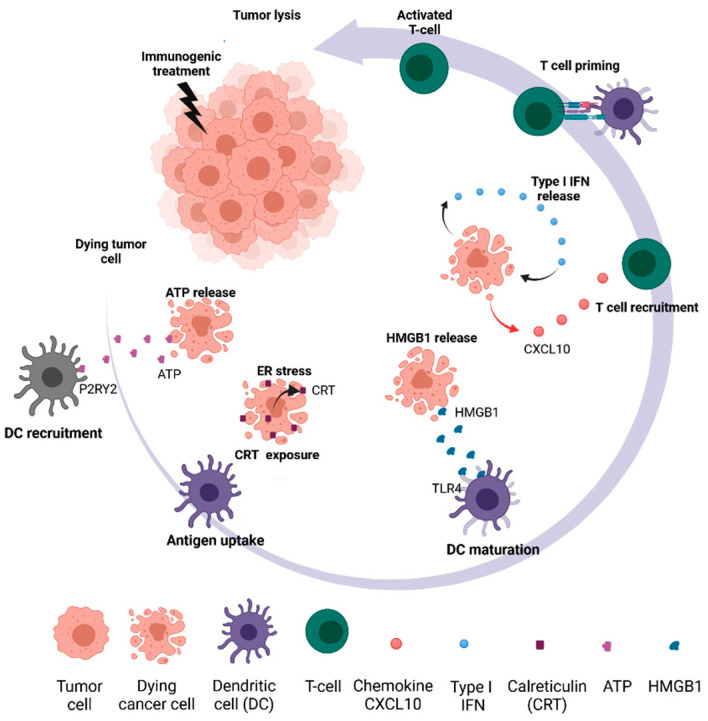
Molecular basis of immunogenic cancer cell death (ICD) (Inspired by Galluzzi et al. 2016 [59] and created with BioRender.com).

**Figure 2 cancers-13-05999-f002:**
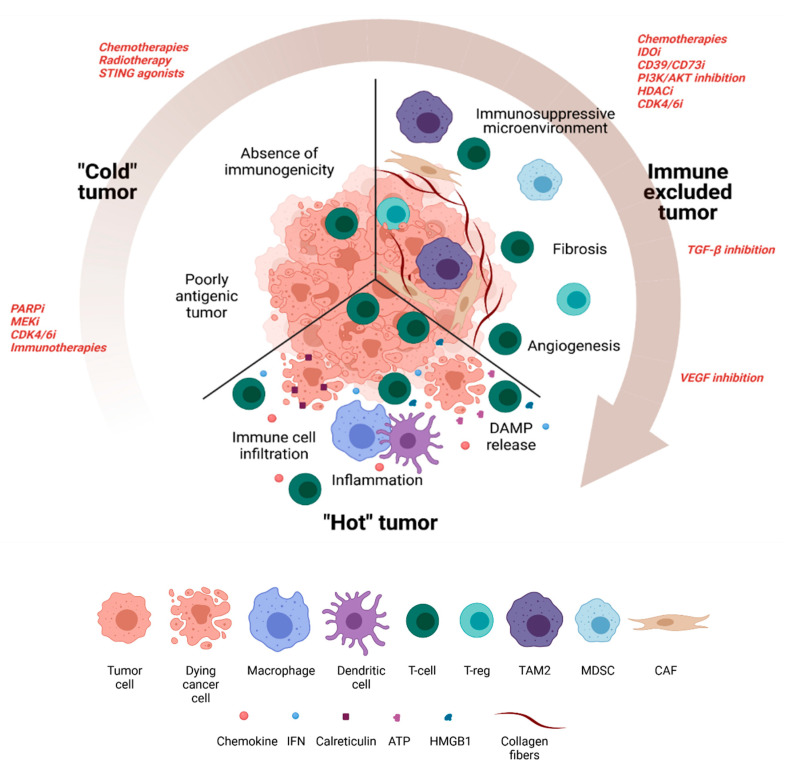
The different tumor immunophenotypes in breast cancer: “cold” tumor, immune-excluded tumor and “hot” tumor (Created with BioRender.com).

**Table 1 cancers-13-05999-t001:** Immunotherapy combinations in clinical trial including anti-PD-(L)1 antibody.

Trial Name/Number	Phase	Subtype/Condition	Immunotherapy Combinations
NCT03219268	I	HER2/TNBC Advanced Solid Tumors	MGD013:Bispecific antibodies anti-LAG3/anti-PD-1
NCT03440437	I/II	Advanced Cancer Metastatic Cancer	FS118: Bispecific antibodies anti-LAG3/anti-PD-L1
DUET-2 NCT03517488	I	Advanced Solid Tumors	XmAb®20717: Bispecific antibodies anti-CTLA4/anti-PD-1
NCT01968109	I/2a	Advanced Solid Tumors	Nivolumab (anti-PD-1 antibody) + BMS-936558 (anti-LAG3 antibody)
AIPAC-002 NCT04252768	I	Metastatic Breast Cancer HR+	Eftilagimod Alpha (Soluble LAG-3 Protein) + paclitaxel
InCITe NCT03971409	II	Stage IV or Unresectable TNBC	PF-04518600 (Anti-OX40 antibody) + Avelumab (anti-PD-L1 antibody)
			4-1BB/CD137 agonist+ avelumab
			Sacituzumab govitecan (anti-Trop2 antibody) + avelumab
NCT02794571	Ia/Ib	Advanced/Metastatic Tumors	Tiragolumab (anti-TIGIT antibody) + Atezolizumab (anti-PD-L1 antibody) + chemotherapies
NCT04584112	Ib	TNBC	Tiragolumab + Atezolizumab + chemotherapies
NCT03742349	I	TNBC	Spartalizumab (anti-PD-1 antibody) + LAG525(anti-LAG-3 antibody) + NIR178 or capmatinib or MCS110 or canakinumab (anti IL-1β antibody)

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
