# Peer review of "Therapeutic Associations Comprising Anti-PD-1/PD-L1 in Breast Cancer: Clinical Challenges and Perspectives"

_cancers, 2021, doi:10.3390/cancers13235999_

Round 1

Reviewer 1 Report

The article entitled "therapeutic associations comprising anti-PD-1/PD-L1 in breast 2 cancer: clinical challenges and perspectives" presents an overview of current advances in developing immunotherapeutics for breast cancer treatments. This is a well-written article. The visibility of this article could be enhanced by clarifying or simplifying certain areas of the review article.

Minor suggestions:

  1. Under section 2.1.4., authors should consider putting a table demonstrating the different combinatorial approach that works with anti-PD-1/PD-L1 therapy.
  2. Figure 1 needs to be simplified and labeled. In the current form, it is hard to understand which cells are doing what and also creates confusion for this reviewer.
  3. Authors should pay attention and provide references for sentences. For example, reference is needed for lines 632 and 633. Similarly, there are missing references throughout the manuscript. 

Reviewer 2 Report

This review by Ledys et al is a fairly comprehensive review of the current situation regarding the application of immunotherapy to breast cancer.  The topic is of considerable importance and the review timely.   It is well written, clear and concise.
There are some minor improvements that could be made:
1.      Consideration of the suggestion that tumor cell expressed PD-1 may play a role in the response in some tumor types (e.g. Tumor cell-intrinsic PD-1 receptor is a tumor suppressor and mediates resistance to PD-1 blockade therapy,   Xiaodong Wang et al).
This may turn out to be an important complicating factor that is not well understood.
  1. A little more discussion of the mechanism of “hot” vs ‘cold”.
  2. her2 negative luminal make up the overwhelming majority..” needs a reference.
  3. Lines 263/264 would benefit from references.
  4. There is a spelling mistake on line 159.

Reviewer 3 Report

The author demonstrates “Therapeutic associations comprising anti-PD-1/PD-L1 in breast 2 cancer: clinical challenges and perspectives” Manuscript is well written and discussed each point sequentially. Manuscript will be suitable for publication once the following point have been addressed.

  1. In the line number 46-47.

here Author should explain the HER-2 positive and negative genetic makeup and protein expression profile. Author should also specify the relative prognosis of different genetic mutation.

  1. Line number 54

Author should first put the full name and then abbreviation (cytotoxic T-lymphocyte-associated protein 4 (CTLA-4)

  1. Line number 57

Author should add the reference here.

  1. Line number 60

Please use latest reference here.

  1. Line number 76 to 78

here Author should also discuss the relative immune infiltration in HER-2 positive and TNBC.

  1. Author should also discuss the cancer stem cells and how immune infiltration in tumor microenvironment affect the cancer stem cells.
  2. Cold tumor (tumor with immune suppression) where immunotherapy is not responding therefor different therapeutic approach should be used. Inhibition of Autophagy in Cold tumor potentiate the immunogenic response in cold tumor and increase the cells death of cancer stem cells (Ref PMID: 32892693 and PMID: 33946505. I will recommend that author should also discussed this point in section 3 (Line number 509)
